

# Individual differences show that only some bats can cope with noise-induced masking and distraction

Dylan G.E. Gomes[1,2] and Holger R. Goerlitz[1]

[1] Max Plank Institute for Ornithology, Acoustic and Functional Ecology, Seewiesen, Germany
[2] Department of Biological Sciences, Boise State University, Boise, ID, United States of America

## ABSTRACT

Anthropogenic noise is a widespread pollutant that has received considerable recent attention. While alarming effects on wildlife have been documented, we have limited understanding of the perceptual mechanisms of noise disturbance, which are required to understand potential mitigation measures. Likewise, individual differences in response to noise (especially via perceptual mechanisms) are likely widespread, but lacking in empirical data. Here we use the echolocating bat *Phyllostomus discolor*, a trained discrimination task, and experimental noise playback to explicitly test perceptual mechanisms of noise disturbance. We demonstrate high individual variability in response to noise treatments and evidence for multiple perceptual mechanisms. Additionally, we highlight that only some individuals were able to cope with noise, while others were not. We tested for changes in echolocation call duration, amplitude, and peak frequency as possible ways of coping with noise. Although all bats strongly increased call amplitude and showed additional minor changes in call duration and frequency, these changes could not explain the differences in coping and non-coping individuals. Our understanding of noise disturbance needs to become more mechanistic and individualistic as research knowledge is transformed into policy changes and conservation action.

## INTRODUCTION

Anthropogenic noise is a global pollutant that is pervasive even in protected areas (*Buxton et al., 2017*), is increasing over time (*Luther & Baptista, 2009*), and has recently gained considerable attention by behavioral biologists (*Barber, Crooks & Fristrup, 2010*). Noise can disrupt animal behavior, such as communication (*Brumm & Slabbekoorn, 2005*; *Rabin et al., 2003*) and foraging (*Gomes et al., 2016*; *Purser & Radford, 2011*; *Siemers & Schaub, 2011*), reduce reproductive success (*Halfwerk, Holleman & Slabbekoorn, 2011*), increase mortality (*Simpson et al., 2016*), change biological communities (*Francis, Ortega & Cruz, 2011*), and alter ecological services (*Francis et al., 2012*). Yet it is not often understood what mechanisms drive these changes, and if and how different individuals are affected by these mechanisms differently. Individual differences in response to noise has been

Corresponding author
Dylan G.E. Gomes,
dylangomes@u.boisestate.edu

documented in humans (*Furnham & Strbac, 2002*; *Standing, Lynn & Moxness, 1990*), birds (*Naguib et al., 2013*), fish (*Bruintjes & Radford, 2013*), mongooses (*Eastcott et al., 2020*), and bats (*Luo, Siemers & Koselj, 2015*; *Luo et al., 2015*; *Simmons et al., 2018*), among many others (reviewed in *Harding et al., 2019*), yet individual differences are often overlooked as individuals are more often grouped together for analysis than analyzed individually (*Harding et al., 2019*).

Similarly, only a few studies to date have investigated the mechanisms of noise disturbance. Some have used bands of noise that are both overlapping or non-overlapping with signals or cues of interest to demonstrate evidence of masking (e.g., *Gomes et al., 2016*; *Zhou, Radford & Magrath, 2019*). Others have shown that noise can disrupt behaviors across sensory modalities (where masking cannot work) via distraction (e.g., *Chan et al., 2010*; *Morris-Drake, Kern & Radford, 2016*). Understanding how we may be able to mitigate the consequences of noise relies heavily on knowledge of direct mechanisms of noise disturbance on individuals. *Dominoni et al. (2020)*, for example, highlight three main perceptual mechanisms of noise disturbance—masking, distraction, and misleading. While these mechanisms apply to all senses, we here consider them specifically in the auditory domain.

Masking is a mechanism whereby noise overlaps in frequency with important signals or cues, thus making the detection and auditory analysis of the signal difficult, if not impossible, and occurs at the auditory periphery (*Clark et al., 2009*; *Fay, 1988*; *Gomes et al., 2016*; *Tanner Jr, 1958*). Distraction, on the other hand, occurs when noise competes for the finite attention (i.e., central processing) of an organism, and is not limited to frequencies that overlap with a signal or cue of interest (*Chan et al., 2010*). Misleading occurs when noise is interpreted as something that it is not, similar to a false alarm (*Wiley, 2013*) or a sensory trap (*Christy, 1995*; *West-Eberhard, 1984*). Some have shown, for example, that beaked whales respond to Navy sonar similarly as they do killer whales, perhaps indicating that they misinterpret this noise as a predator or something unknown that might be dangerous (*Tyack et al., 2011*). Other mechanisms of disturbance have been proposed, such as stress, fear, and avoidance (*Campo, Gil & Davila, 2005*; *Luo, Siemers & Koselj, 2015*; *Voellmy et al., 2014*), yet these physiological and behavioral responses must occur downstream of the initial perceptual mechanism (i.e., masking, distraction, or misleading).

Here, we use a behavioral experiment to tease apart the effects of both masking and distraction as perceptual mechanisms on individual echolocating bats. Echolocating bats are a worthwhile system to study these questions because they actively sense their world via sound. Thus, by monitoring how they adjust the characteristics of their echolocation calls in response to noise, we can easily study how they are responding to changes in the sensory environment.

Anthropogenic noise arises from diverse sources (e.g., automobile and airplane traffic, energy extraction, and urban cities) and generally contains more energy in lower rather than in higher frequencies (*Bondello & Brattstrom, 1978*; *Bunkley & Barber, 2015*; *Bunkley et al., 2015*; *Cinto Mejia, McClure & Barber, 2019*; *Nemeth & Brumm, 2010*; *Schaub, Ostwald & Siemers, 2008*; *Siemers & Schaub, 2011*; *Sierro et al., 2017*). Despite this, anthropogenic noise can contain considerable energy in the ultrasonic range, particularly at close distance

to the noise source (*Siemers & Schaub, 2011*). As the echolocation call frequencies of different bat species also span a wide frequency range (*Fenton & Bell, 1981*), anthropogenic noise can overlap bat echolocation in frequency in a species-, distance- and noise-source specific manner, potentially affecting bats via multiple mechanisms (*Dominoni et al., 2020*).

We trained pale spear-nosed bats (*Phyllostomus discolor*) to discriminate surface structures with increasing level of difficulty and under three noise treatments (see Methods). We made distinct predictions for each of the tested perceptual mechanisms. By broadcasting noise that does and does not spectrally overlap with echolocation calls, we directly tested the role of masking. We predicted that masking should only reduce the discrimination performance for spectrally overlapping noise, but not for non-overlapping noise. Since distraction assumes that deficits result from limited attentional resources, we predicted that distraction should be independent of the noises' spectral overlap with echolocation calls (distinguishing it from masking), but should depend on the noises' temporal structure. We thus also presented a spectrally overlapping 'sparse' noise with random temporal gaps, making the noise less predictable, and thus, more distracting (*Glass & Singer, 1972*; *Kjellberg et al., 1996*; *Matthews et al., 1980*). At the same time, sparse noise might allow bats to listen in-between the noise gaps ("dip listening"), reducing its masking effect (*Vélez & Bee, 2011*). Thus, if distraction is the primary mechanism of disturbance, then sparse noise should decrease discrimination performance and increase trial duration, while we would expect an increase in discrimination performance and a decrease in trial duration under sparse noise if masking is the primary mechanism of disturbance.

## MATERIALS AND METHODS

### Experimental animals and husbandry

The pale spear-nosed bat (*Phyllostomus discolor*; Wagner, 1843) is an omnivorous neotropical bat (*Kwiecinski, 2006*) that emits multi-harmonic, downward frequency-modulated echolocation calls of short duration (0.3–2.5 ms) and most energy in the range of ~40–100 kHz (*Goerlitz, Hübner & Wiegrebe, 2008*; *Kwiecinski, 2006*). A captive colony of *P. discolor* was kept in a temperature (~25 °C) and humidity (~70%) controlled room at the Max Planck Institute for Ornithology, Seewiesen, Germany, where the bats had access to water *ad libitum*, and were fed a fruit-based diet. During experimental days, bats were first only fed during experiments (mealworm reward; *see below*), to maintain motivation. At the end of the day, several hours later, bats were fed fruit. Experiments were carried out in a nearby, but separate room (~21 °C / 65% rel. hum). Bat housing and all research was approved by the German authorities under the permit numbers 311.5–5682.1/1-2014-023 (Landratsamt Starnberg) and 55.2-1-54-2532-18-15 (Regierung von Oberbayern), respectively.

### Experimental setup

Experiments were conducted in a dark chamber within a dark room (see below for light levels). Walls of both the chamber and the room were covered in anechoic foam to reduce echoes. The chamber held a custom-built mushroom maze (87 cm × 65 cm × 18 cm, W × H × D), which was a fully enclosed mesh-cage (*Baier, Wiegrebe & Goerlitz, 2019*).

The mesh (12 mm × 12 mm mesh grid with wires of <1 mm diameter) is acoustically transparent for the echolocation calls of P. discolor (wavelength: ∼3–9 mm). This maze allowed the bats perceptual access by echolocation to two simultaneously presented stimulus discs (reference plus test disc) on either side of the maze (Fig. 1A). One infrared light barrier next to each of the disc positions objectively recorded the choice of the bat via a custom-written Matlab code (The Mathworks, Nattick, MA, USA), avoiding observer bias and potential observer errors. Two loudspeakers (XT25SC90-04, Tymphany, San Rafel, CA, USA; connected to power amplifier TA-FE330R, Sony, Tokyo, Japan, and soundcard Fireface 802, RME, Haimhausen, Germany) were mounted on either side of the setup for noise playback (Fig. 1A). The experimenter (stationed outside of the chamber) observed the experiment via a red-filtered computer screen displaying a live-feed from an infrared camera (Foculus FO432SB; NET-GmbH, Finning, Germany; 880 nm infrared LED-illumination, TV6818; ABUS, Wetter, Germany).

## Stimuli

We used an established behavioral assay that has been previously used to test perceptual performance in bats (*Baier, Wiegrebe & Goerlitz, 2019*). We used eight discs with 45 cm diameter as physical stimuli. The stimulus discs were made by a milling cutter (Modellbau Grossmann, Calw, Germany) and then spray-painted with multiple coats to be smooth-textured. One disc ("reference disc") had a completely flat surface. The seven other discs had concentric ripples, resembling concentric sinusoidal standing waves. All rippled discs had the same spatial frequency of 17.8 ripples per meter, corresponding to eight full sinusoidal ripples per disc, but different ripple heights increasing from 2 to 32 mm peak-to-peak height (2, 4, 5.6, 8, 11.2, 16, 32 mm; Fig. 1B).

## Noise treatments

In addition to silence, used as a control, we tested the bats under three band-limited white noise treatments (Fig. 2): (1) Smooth non-overlapping noise: band-limited white Gaussian noise not overlapping in frequency with the echolocation calls of *P. discolor*, ranging from 5–35 kHz (10th-order Butterworth filter). (2) Smooth-overlapping noise: band-limited white Gaussian noise overlapping in frequency with the echolocation calls of *P. discolor*, ranging from 40–90 kHz (10th-order Butterworth filter). (3) Sparse-overlapping noise: derived from the smooth-overlapping noise (40–90 kHz) with additional short silent gaps between all adjacent samples (*Hübner & Wiegrebe, 2003*). These silent gaps generate fluctuations in the temporal envelope of the noise, causing the noise to sound rougher in comparison to the smooth noise. The duration of the silent gaps was drawn from a uniformly random duration (mean: 0.3 ms, range: 0-0.6 ms). The roughness is quantified by the base-10 logarithm of the waveform's fourth moment LogM4; (*Hartmann & Pumplin, 1988*), and is calculated as the summed amplitude values raised to the power of 4, divided the squared sum of all squared amplitude values. LogM4 was 1.44 logM4 for the sparse-overlapping noise, compared to 0.48 logM4 for the two smooth noises (cf. *Grunwald, Schörnich & Wiegrebe, 2004*). For all three noise types, we generated uncorrelated stereo noise files of 60 min duration (192 kHz sampling rate, 16 bit resolution) and corrected each

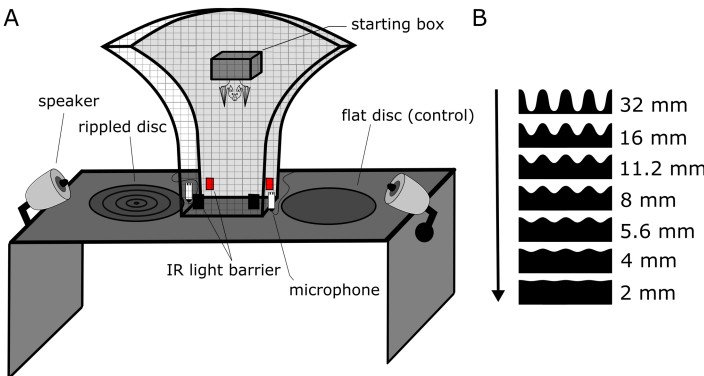

**Figure 1** **Experimental setup and stimuli of the discrimination experiment.** (A) Sketch of the experimental setup. The bat could freely move within the mushroom-shaped mesh-cage, allowing simultaneous perceptual access to both discs from multiple angles. Each trial started when the bat left from within the starting box and ended when the bat crossed an infrared light barrier next to each disc, objectively determining decision and duration of each trial. The bat received a food reward for approaching the flat reference disk. Noise treatments were presented via two speakers from similar directions as the returning disc echoes. Echolocation calls were recorded via microphones next to the light barriers. (B) Cross-sections of the stimulus discs. The peak-to-peak height of the ripples is scaled to size and given on the right (but the shown distance between ripple peaks is shorter, i.e., higher spatial frequency, than in the experiment for better visualization). As ripple height gets smaller, the task to discriminate the rippled disk from the flat reference disc becomes more difficult, as indicated by the arrow.

channel for the corresponding speaker's frequency response. At each daily experimental session, noise playback was started at least 30 s prior to the beginning of the first trial at a random position within the 60-min long noise files, and continued throughout the session. Playback level was 70 dB SPL RMS re. 20 µPa at the bat's starting position. It is important to note that the perceived loudness was likely different, since the noise treatments had different bandwidths and the auditory sensitivity of *P. discolor* varies over their range of hearing (*Esser & Daucher, 1996*; *Hoffmann et al., 2008*). However, this should not affect any interpretation of the designed tests of masking and distraction.

## Training and testing

Nine bats were initially trained in a two-alternative forced-choice paradigm to discriminate the flat reference disc from the stimulus disc with the highest ripples (32 mm). During training, bats received mealworms (larvae of *Tenebrio molitor*) as reward when approaching the flat reference disc only. Two bats did not learn this task at all, and three bats appeared to learn somewhat but never showed consistent performance after three months of training (including one bat that had already been trained successfully in a previous experiment; *Baier, Wiegrebe & Goerlitz, 2019*). There appears to be no consistent patterns between failure to train and previous experience or age of the bats. Only four bats (3 males, 1 female) reached the training criterion (>70% correct approaches to the flat reference disc over three consecutive days; the three males were previously trained in *Baier, Wiegrebe & Goerlitz, 2019*) and were subsequently tested during the data acquisition stage. Throughout testing and training, the flat reference disc was pseudo-randomly (*Gellermann, 1933*)
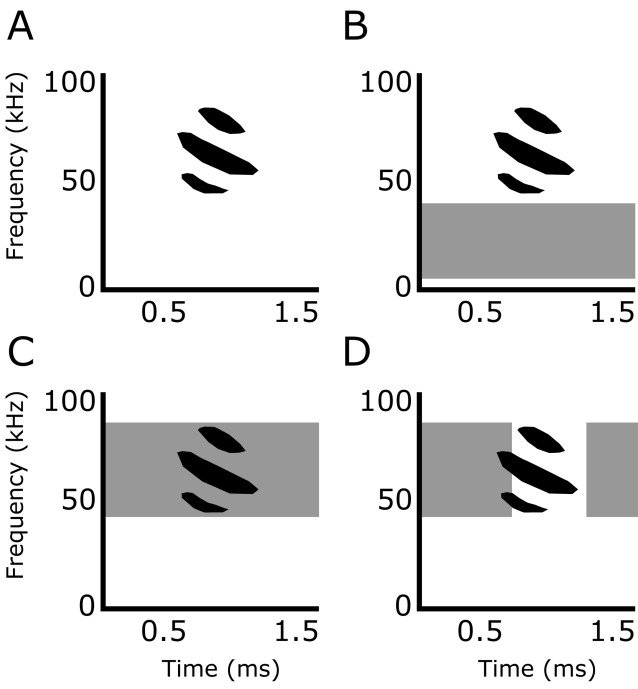

**Figure 2 Experimental playback treatments.** Representative echolocation calls of *Phyllostomus discolor* in (A) silence (control), (B) smooth non-overlapping noise (5–35 kHz), (C) smooth overlapping noise (40–90 kHz), and (D) sparse overlapping noise (40–90 kHz with additional silent gaps of variable duration introduced between all adjacent noise samples; the duration of the silent gaps was drawn from a uniform distribution that ranged from 0 to 0.6 ms and had a mean value of 0.3 ms). Echolocation call is based on *Goerlitz, Hübner & Wiegrebe (2008)*.

alternated between each side of the experimental setup to avoid any biases in location preferences by the bats.

Prior to each trial, bats were encouraged to enter a small Tupperware 'starting' box in the middle of the experimental setup by offering blended banana as a food reward via a syringe tube that was mounted inside this starting box (Fig. 1A). While in the starting box, bats had no perceptual access to the discs since the solid bottom door was closed upon entering. The discs were then placed in their positions. Once the starting box opened, the trial started. Bats were allowed to crawl through the setup towards the discs. When they broke the IR light barrier, the trial ended, and the bat was rewarded with a mealworm when it chose the flat reference disc.

Bats were initially tested in a silent (i.e., ambient sound level) experimental room to generate baseline psychometric curves with discs of 5 different ripple heights (two mm, four mm, eight mm, 16 mm, and 32 mm). For the subsequent tests, we added two additional discs with intermediate ripple heights (5.6 mm and 11.2 mm) to get better resolution around the turning point of the psychometric function measured in silence. The bats were then tested with all seven discs in each of the three noise conditions. Per individual, the noise conditions were presented in pseudo-random sequence, and a new noise condition was presented only after the individual had completed all trials of the previous noise

condition. Finally, each bat was retested a second time in silence with all seven discs, to assure that differences in performance were not due to learning or other order effects. Testing each bat in the same noise conditions over multiple trials and days (as opposed to changing noise treatment for each trial) may lead to more habituation to that noise treatment. However, as the opportunity for habituation was consistent across treatments, any differences amongst noise treatments are due to the noise treatments themselves. Additionally, as randomizing noise treatments across trials would only exacerbate any differences amongst noise treatments, the differences we report here are conservative. Each bat was tested 30 times for every ripple height and noise treatment combination, totaling 990 trials per individual bat.

To motivate the bats, each day was started with easier discrimination tasks (higher ripple heights) and gradually moved towards more difficult tasks (lower ripple heights). Bats were allowed to continue testing until satiated or no longer food-motivated, which was determined by the bat attempting to leave the mushroom maze via an exit door in the top of the setup.

## Aborted trials

If the bat did not exit the starting box within 5 min after starting a trial, the trial was aborted and repeated. As the bats did not make a decision in those aborted trials, they were not included in further analyses. The one exception, however, is that we analyzed the number of these aborted trials as a measure of aversion to the noise. Abortion was behaviorally distinguished from satiation, as bats would crawl toward the door to exit the maze when they were seemingly no longer food-motivated. It is likely that bats aborted trials not only because of the noise, but also for potential other reasons. However, as those other reasons are constant within the controlled setting of an experiment, any *difference* in the number of aborted trials (as in any other behavioral measure) can be attributed to the experimental treatment (i.e., the type of noise).

## Echolocation call recording and analysis

We continuously recorded the bats' echolocation calls into a 4-sec-long ring-buffer. When a bat made a decision by interrupting a light barrier, only the sound of the four seconds prior to the decision was automatically saved into a WAV-file. Recordings were performed with two microphones (Knowles SPU0410) positioned just behind each light barrier, a sound card (Fireface 802, RME, Haimhausen, Germany; 192 kHz sampling rate, 16-bit resolution) and playrec (V2.1.0, playrec.co.uk) for Matlab (V2007b, The Mathworks, Nattick, MA, USA).

Echolocation calls were analyzed automatically by custom-written scripts in Matlab (V2016a, The Mathworks, Nattick, MA, USA), advanced from previous work (*Goerlitz, Hübner & Wiegrebe, 2008*; *Luo, Siemers & Koselj, 2015*). First, we filtered all recordings with each microphone's compensatory impulse response (511-order finite impulse response filter) to compensate for the microphone's frequency response, and a band-pass filter (38–95 kHz, 8th-order elliptic filter). Second, we used a threshold detector to broadly determine the timing of all acoustic events: we additionally band-pass-filtered recordings

around the bats' main call energy (45–90 kHz, 4th-order elliptic filter), calculated their low-pass filtered (500 Hz, 4th-order elliptic filter) Hilbert-envelope, and detected all acoustic events that surpassed a threshold (mean + 2x STD of the envelope), excluding events that were too close to the preceding event (<20 ms) and too short (<0.75 ms). We then added an additional 0.5 ms on both sides of the detected acoustic events, which, together with the previous low-pass-filtering of the envelope, ensured that the determined time window included the full call flanked by non-call samples. Third, we detected the actual call within this time window of the recording, and analyzed its acoustic parameters.

Call duration was determined from the low-pass filtered (5000 Hz, 2nd-order butterworth filter) Hilbert envelope of the originally filtered recording (38–95 kHz) at -12 dB below the envelope's peak value. Peak frequency (frequency with highest amplitude), frequency centroid (dividing the call energy into two halves along the frequency axis; *Au, 2012*) and the lowest and highest frequency (defined as the lowest and highest frequencies whose amplitudes were at −12 dB below the highest amplitude) were calculated from the time-averaged call spectrogram (1024 FFT of 100 samples, 95% overlap). Relative call level was calculated as the root mean square (RMS) of all samples within the −12 dB duration criterion and expressed in dB FS, i.e., negative dB values relative to the full scale of the recording system.

If a call was detected on both microphones, we only analyzed the call with the higher signal-to-noise-ratio (SNR: call-RMS relative to RMS of all parts of the recording that were not classified as acoustic events). Of all recorded calls ($N = 287{,}061$), we excluded for further analysis calls shorter than 0.3 ms and longer than 2 ms, with too high (>−0.5 dB FS, to avoid clipping) or too low recorded peak amplitudes (<−15 dB FS), with a SNR of less than 20 dB, and whose ratio between the −12 dB duration and the -6 dB duration was larger than 1.5 (to exclude calls with long echoes). All remaining calls ($N = 63{,}990$) were manually viewed as spectrogram (256 FFT, 50 time slices over full call length, 95% overlap), blind to bat individual and noise treatment, to exclude ambiguous recordings and obvious artefacts, e.g., overlapping call-echo-pairs and non-multiharmonic sounds (e.g., clicks, external noise). This resulted in a final data set of 59,173 calls (0-83 calls per trial). Of the 3960 total trials, 3469 trials (87.6%) included at least one recorded call, 3166 trials (80.0%) at least 3 calls, 2912 trials (73.5%) at least 5 calls, and 2265 trials (57.2%) included at least 10 recorded calls. For further analysis, we used the mean call parameters of each trial in statistical models. Note that the background noise affects our call level measurements only negligibly (<0.05 dB) because we only analyzed calls with a SNR >20 dB.

## Visual system and light levels

Light levels in the experimental room were extremely low ($1.39 \times 10^{-5}$ lux; SPM068 with ILT1700 light detector, resolution $10^{-7}$ lux, International Light Technologies, Peabody, MA, USA), precluding the use of vision to discriminate between discs. Many other laboratory experiments, which have similarly excluded the use of vision due to an assumed unavailability of light, have either reported higher light levels than us or did not measure or report light levels. Additionally, it has been experimentally shown that another related phyllostomid bat (*Macrotus californicus*) only has visual acuity to light levels as low as

$2 \times 10^{-3}$ lux (*Bell & Fenton, 1986*), which is nearly two orders of magnitude higher than our light levels. Furthermore, *M. californicus* has one of the highest sensitivities to low light levels known (*Bell & Fenton, 1986*; *Eklöf et al., 2014*). Thus, it is extremely unlikely that the *Phyllostomus discolor* used here were able to visually discriminate between the discs.

## Statistical analysis

We fitted (generalized) linear models to the behavioral data of each individual, using R (*R Core Team, 2017*). Response variables were analyzed with different distribution families and link functions based on theoretical sampling distributions of the data, and model fits were validated with plots of model residuals, and were checked for collinearity.

We used a binomial distribution family and logit link function to analyze differences in discrimination performance and number of aborted trials, since these were binary data. We used an inverse Gaussian distribution family with an identity link function to analyze trial time data (*Baayen & Milin, 2010*). Log-normal linear models (Gaussian family with an identity function) were used to analyze log-transformed received call level, duration, peak frequency, and frequency centroid. Peak frequency and frequency centroid measure similar aspects of vocalization frequency and are both used in the literature (*Goerlitz, Hübner & Wiegrebe, 2008*; *Holderied et al., 2005*; *Lattenkamp, Vernes & Wiegrebe, 2018*; *Lazure & Fenton, 2011*). We therefore included both metrics in our analyses for comparability, but report only peak frequency in the main text because it is the most commonly used metric, and present frequency centroid data in the supplementary information (Table S6).

We initially checked that discrimination performance did not change between the first and last (pre- and post-treatment) silent conditions with a logistic regression including noise treatment (as factors) and ripple height. Since the performance across the two silent noise conditions did not change for any of the bats (see Results), we pooled these data for analysis, and thus present both 'silent' conditions together.

For all models we used noise treatment, ripple height, and their interaction as explanatory variables, while the number of days that bats were in our experiment was included as a covariate, all fitted as fixed effects. The number of days an animal is in an experiment may influence performance because animals may learn over time, become faster at a given task, or, conversely, give up on difficult tasks. We thus included this term as a covariate in our model to control for it (and show its effect in the model output tables), but focused our main analysis and interpretation on more pertinent variables.

We fitted individual models for each bat, instead of single models for every response variable, with bats as random effects terms (e.g., *Gomes et al., 2017*), for two reasons. Firstly, it has been suggested that random effects terms should have a minimum of five groups; otherwise estimates of variance become imprecise (*Harrison et al., 2018*). As we only had four bats that completed the experiment, we were unable to fulfill this requirement. Secondly, and more importantly, fitting models to each individual bat allowed us to understand the nuanced differences between them, which an all-bats-combined model would not achieve. Since we fitted four models per response variable (one for each bat), we used conservative Bonferroni corrections to correct *p* values for these multiple comparisons

by multiplying $p$ values by four. All differences reported in results due to noise treatments are model estimates, and not differences in raw data.

### Performance thresholds

We used a binomial generalized linear model with a probit link (constrained between 0.5 and 1 with the link function 'mafc.probit' in the R package 'psyphy'; *Knoblauch, 2007*) to generate estimates of ripple height thresholds at which bats exceed correct responses at least 70% of the time. For each bat, 1000 simulated discrimination performance (0 or 1) datasets were generated based on the above model estimates for each bat, at each ripple height, within each noise treatment. Then the lower 0.025 and upper 0.975 percentiles of those data gave us a 95% confidence interval band around our performance threshold.

## RESULTS

### Discrimination performance

All four bats learned to discriminate the smooth disc from the rippled disc with the highest ripples (32 mm) in silence (88–100% correct), and showed reduced discrimination performance with decreasing ripple height (Fig. 3, orange line; logistic model $p < 0.001$; Table 1). In silence, performance dropped below our 70% threshold criterion for ripple heights around 7.9 mm (mean; range = 5.4–11.4 mm; Fig. 3; Table 2), matching the mean threshold found by *Baier, Wiegrebe & Goerlitz (2019)* of 8.0 mm (range: 3.7–12.3 mm). As the performance in silence did not change for any bat between the first and last silence condition (Bat A: $p = 0.21$, B: $p = 0.89$, C: $p = 0.81$, D: $p = 0.39$), demonstrating a lack of order (e.g., learning) effects, we pooled all silence trials. Noise treatments did not change the discrimination performance of bats A and B (hereafter 'coping' bats; Table 1), and the 95% confidence intervals of their thresholds in noise overlapped with those in silence. In contrast, discrimination performance decreased for bats C and D (hereafter 'non-coping' bats) both under smooth-overlapping noise ($z = -3.2$, $p = 0.008$; $z = -3.2$, $p = 0.004$) and sparse-overlapping noise ($z = -3.7$, $p < 0.001$; $z = -3.1$, $p = 0.008$; indicated as blue and purple lines in Fig. 3 respectively). The same is true for the smooth non-overlapping noise for bat C ($z = -2.5$, $p = 0.047$, green line), yet not for bat D ($z = 1.8$, $p = 0.24$).

### Trial duration

The time to complete trials differed between some noise treatments for some bats (Fig. 4). Both bats A and D made faster decisions during smooth-overlapping noise compared to silence (model estimated trial durations of bat A and D under noise and silence, respectively: 28.8 s vs. 30.0 s (A) and 10.3 s vs. 14.8 s (D); $z = -3.4$, $p = 0.004$ (A); $z = -4$, $p < 0.001$ (D); Table S1). However, bat C took longer to complete trials during sparse-overlapping noise (48.6 s vs. 18.5 s; $z = 4.8$, $p < 0.001$), while noise treatments did not affect the trial time of bat B (Table S1).

### Aborted trials

The bats aborted 297 trials of 4,257 total trials (7%; bats A: 54; B: 92; C: 101; D: 50). Compared to silence, both bats B and C significantly aborted more trials under both

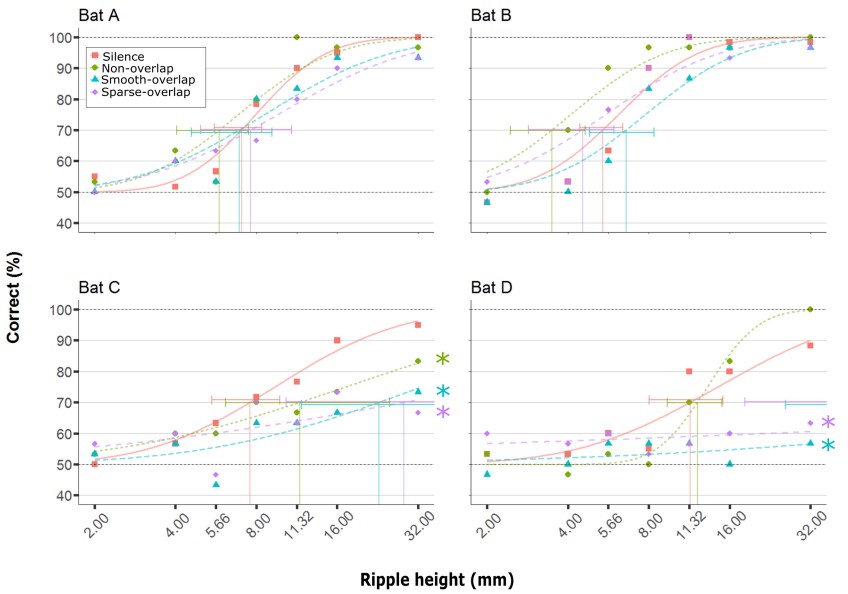

**Figure 3 Discrimination performance of four bats as a function of peak-to-peak disc ripple height during silent control and three noise treatments.** Shown are raw data (dots) and the fitted performance curve (colored lines; constrained probit function) for each noise treatment. The discrimination threshold is indicated by the vertical line, with its 95% confidence interval. Discrimination performance of 'coping' bats (A and B) did not differ between silence and noise treatments. In contrast, discrimination performance of 'non-coping' bats (C and D) was reduced under both overlapping noise treatments (blue and purple lines), while discrimination performance of bat C was also reduced in non-overlapping smooth noise (green line). Noise treatments have been abbreviated here (as compared to the text) to reduce visual clutter (smooth non-overlapping noise = "Non-overlap"; smooth-overlapping noise = "Smooth-overlap"; sparse-overlapping noise = "Sparse-overlap"). Asterisks denote that the interaction between noise treatment and ripple height differs significantly ($p < 0.05$) from the silent controls (orange lines) in generalized linear models.

smooth non-overlapping ($z = 4.0$, $p < 0.001$; $z = 3.0$, $p = 0.01$) and smooth-overlapping noise ($z = 3.9$, $p < 0.001$; $z = 4.2$, $p < 0.001$). In addition, bat B and also bat D aborted more trials under sparse-overlapping noise compared to silence ($z = 5.2$, $p < 0.001$; $z = 3.8$, $p < 0.001$; Fig. 5; Table S2).

## Echolocation call parameters

We recorded at least one call for most (87.6%) of the 3960 total trials, and at least ten calls for more than half of the trials (57.2%), and used the mean call parameters of each trial in the analysis. Mean call duration ranged from 0.38 ms (bat A) to 0.47 ms (bat C), matching previous results in confined space (*Goerlitz, Hübner & Wiegrebe, 2008*; *Luo et al., 2015*). All bats increased call duration under smooth-overlapping noise. Coping bats (A and B) increased call duration by an estimated 0.07 ms, while non-coping bats (C and D) only increased call duration by 0.05 ms and 0.04 ms (bat A: $t = 20.4$, $p < 0.001$; B: $t = 9.1$, $p < 0.001$; C: $t = 7.9$, $p < 0.001$; D: $t = 6.0$, $p < 0.001$; Table S3). Similarly, coping bats increased call duration in sparse-overlapping noise (increase of 0.06 ms and 0.07 ms, A and B respectively), while non-coping bats did not (A: $t = 15.1$, $p < 0.001$; B:

**Table 1  Results from generalized linear models for discrimination performance in various noise treatments.** Model results show the estimated differences in discrimination performance (relative to control trials) for the three noise treatments, ripple height, the number of days the bat was in the experiment, and the interaction between each noise treatment and ripple height (i.e. the shape of each performance curve as a function of ripple height), separately for each bat. Data were analyzed with binomial distribution and logit link function. Noise treatments have been abbreviated here (as compared to the text) to reduce visual clutter (smooth nonoverlapping noise = "Non-overlap"; smooth-overlapping noise = "Smooth-overlap"; sparseoverlapping noise = "Sparse-overlap").

| Bat | Variable | Estimate | SE | Z value | p value |
|-----|----------|----------|-----|---------|---------|
| A | (Intercept) | −0.766 | 0.275 | −2.791 | 0.02 |
| A | Non-overlap | 0.228 | 0.521 | 0.438 | 0.987 |
| A | Smooth-overlap | 0.317 | 0.423 | 0.749 | 0.911 |
| A | Sparse-overlap | 0.544 | 0.447 | 1.216 | 0.637 |
| A | Ripple height | 0.294 | 0.047 | 6.222 | <0.001 |
| A | Day of experiment | 0.110 | 0.095 | 1.165 | 0.673 |
| A | Non-overlap:Ripple height | 0.079 | 0.099 | 0.798 | 0.891 |
| A | Smooth-overlap:Ripple height | −0.096 | 0.064 | −1.504 | 0.435 |
| A | Sparse-overlap:Ripple height | −0.088 | 0.069 | −1.265 | 0.603 |
| B | (Intercept) | −0.544 | 0.243 | −2.240 | 0.096 |
| B | Non-overlap | 0.200 | 0.412 | 0.486 | 0.981 |
| B | Smooth-overlap | 0.441 | 0.377 | 1.171 | 0.668 |
| B | Sparse-overlap | 0.490 | 0.364 | 1.346 | 0.543 |
| B | Ripple height | 0.216 | 0.036 | 6.082 | <0.001 |
| B | Day of experiment | 0.003 | 0.107 | 0.025 | 1 |
| B | Non-overlap:Ripple height | −0.012 | 0.059 | −0.209 | 0.999 |
| B | Smooth-overlap:Ripple height | −0.091 | 0.048 | −1.905 | 0.209 |
| B | Sparse-overlap:Ripple height | −0.109 | 0.045 | −2.405 | 0.062 |
| C | (Intercept) | −0.130 | 0.200 | −0.649 | 0.945 |
| C | Non-overlap | 0.277 | 0.312 | 0.888 | 0.847 |
| C | Smooth-overlap | 0.228 | 0.305 | 0.745 | 0.912 |
| C | Sparse-overlap | 0.433 | 0.298 | 1.451 | 0.471 |
| C | Ripple height | 0.118 | 0.022 | 5.465 | <0.001 |
| C | Day of experiment | −0.071 | 0.087 | −0.817 | 0.882 |
| C | Non-overlap:Ripple height | −0.071 | 0.028 | −2.518 | 0.047 |
| C | Smooth-overlap:Ripple height | −0.085 | 0.027 | −3.157 | 0.008 |
| C | Sparse-overlap:Ripple height | −0.100 | 0.027 | −3.726 | <0.001 |
| D | (Intercept) | −0.069 | 0.177 | −0.387 | 0.992 |
| D | Non-overlap | −0.486 | 0.327 | −1.487 | 0.445 |
| D | Smooth-overlap | 0.118 | 0.278 | 0.422 | 0.989 |
| D | Sparse-overlap | 0.280 | 0.283 | 0.990 | 0.789 |
| D | Ripple height | 0.074 | 0.015 | 4.937 | <0.001 |
| D | Day of experiment | 0.085 | 0.076 | 1.129 | 0.699 |
| D | Non-overlap:Ripple height | 0.060 | 0.033 | 1.832 | 0.242 |
| D | Smooth-overlap:Ripple height | −0.066 | 0.021 | −3.179 | 0.004 |
| D | Sparse-overlap:Ripple height | −0.066 | 0.021 | −3.141 | 0.008 |

**Table 2 Threshold of the discrimination performance for ripple detection.** The threshold is the ripple height where bats exceeded a 0.7 probability of a correct choice. For each bat, 1000 simulated discrimination thresholds were generated with a binomial generalized linear model. The lower 0.025 and upper 0.975 percentiles of those data give lower and upper values of the 95% confidence intervals. Noise treatments have been abbreviated here (as compared to the text) to reduce visual clutter (smooth non-overlapping noise = "Non-overlap"; smooth-overlapping noise = "Smooth-overlap"; sparse-overlapping noise = "Sparse-overlap").

| Noise | p | Threshold | Bat | Lower 95% CI | Upper 95% CI |
|---|---|---|---|---|---|
| Silent (Control) | 0.7 | 7.05 | A | 5.57 | 8.37 |
| Silent (Control) | 0.7 | 5.39 | B | 4.42 | 6.41 |
| Silent (Control) | 0.7 | 7.58 | C | 5.47 | 9.81 |
| Silent (Control) | 0.7 | 11.42 | D | 8.02 | 15.14 |
| Non-overlap | 0.7 | 5.83 | A | 4.05 | 7.49 |
| Non-overlap | 0.7 | 3.49 | B | 2.44 | 4.63 |
| Non-overlap | 0.7 | 11.62 | C | 6.16 | 25.07 |
| Non-overlap | 0.7 | 12.14 | D | 9.37 | 14.98 |
| Smooth-overlap | 0.7 | 6.91 | A | 4.59 | 9.15 |
| Smooth-overlap | 0.7 | 6.60 | B | 4.81 | 8.38 |
| Smooth-overlap | 0.7 | 22.92 | C | 11.77 | 702.42 |
| Smooth-overlap | 0.7 | 621.33 | D | 25.87 | >100000 |
| Sparse-overlap | 0.7 | 7.63 | A | 4.97 | 10.81 |
| Sparse-overlap | 0.7 | 4.54 | B | 2.85 | 5.95 |
| Sparse-overlap | 0.7 | 28.40 | C | 10.32 | >100000 |
| Sparse-overlap | 0.7 | 4395.17 | D | 18.24 | >100000 |

$t = 9.3$, $p < 0.001$; C: $t = 1.2$, $p = 0.22$; D: $t = 1.8$, $p = 0.07$; Fig. 6). Oddly, bats B and C decreased call duration by 0.03 ms and 0.02 ms in non-overlapping noise relative to silence (B: $t = -4.1$, $p < 0.001$; C: $t = -3.8$, $p < 0.001$).

Relative to silence, all bats increased their call sound pressure level in both overlapping noise treatments by about 10–13 dB (smooth overlapping noise: bats A: 11.8 dB; B: 9.7 dB; C: 9.6 dB; D: 13.3 dB ($t = 46.1$; $t = 31.8$; $t = 33.0$; $t = 25.8$); sparse overlapping noise: A: 12.6 dB; B: 8.7 dB; C: 10.5 dB; D: 13.3 dB; ($t = 47.4$; $t = 29.8$; $t = 37.6$; $t = 19.8$), all $p < 0.001$; Fig. 7; Table S4). Additionally, bat A also increased call level during the smooth non-overlapping noise, though by a much lower magnitude of only 1.5 dB ($t = 4.7$, $p < 0.001$).

The mean peak frequency was 69.8 kHz (bats A: 71.8 kHz; B: 69.9 kHz; C: 69.6 kHz; D: 67 kHz). Of all 12 comparisons, only three showed significant, yet small changes of call frequency with no clear pattern: Bat A increased peak frequency in smooth non-overlapping noise by 1.2 kHz, and decreased peak frequency in smooth overlapping noise by 1.4 kHz ($t = 3.8$, $p < 0.001$; $t = -4.5$, $p < 0.001$). Bat C increased peak frequency by 2.1 kHz only in sparse overlapping noise ($t = 5.5$, $p < 0.001$; Fig. 8). Bats B and D never changed their peak frequency (Table S5).

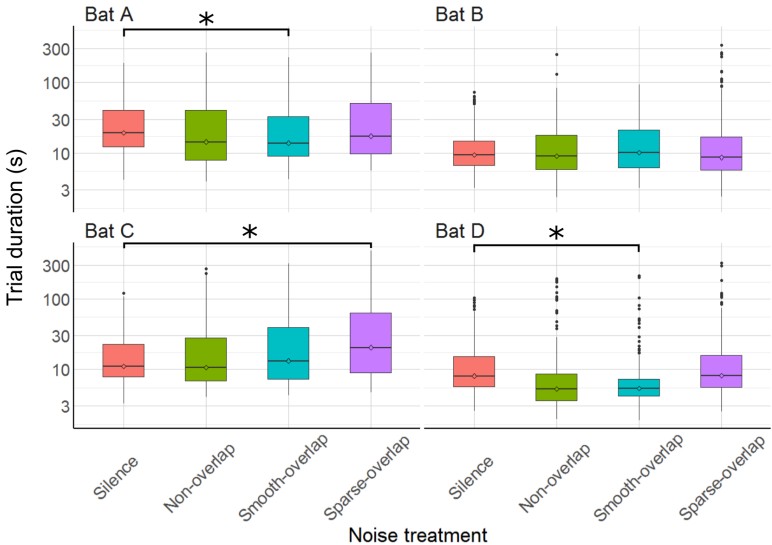

**Figure 4 Trial duration of the discrimination task by noise treatment.** Smooth-overlapping noise re-duced trial duration of bats A and D, and sparse-overlapping noise increased trial duration of bat C. Box plots show median (solid line) and first and third quartiles (edges of box); whiskers extend to the rest of the data minus outliers (more than 1.5 times the interquartile range), which are shown as points. Noise treatments have been abbreviated here (as compared to the text) to reduce visual clutter (smooth non-overlapping noise = "Non-overlap"; smooth-overlapping noise = "Smooth-overlap"; sparse-overlapping noise = "Sparse-overlap"). Asterisks denote significant differences ($p < 0.05$) in trial duration relative to the silence control.

# DISCUSSION

We tested the ability of four bats to discriminate increasingly rippled surface structures from a flat surface under silence and three different noise types. By comparing the bats' discrimination performance, behavior, and echolocation parameters, we address the perceptual mechanism of noise disturbance, and how bats may be able to cope with noise disturbance. The individual bats in our experiments responded to noise in varying ways. Two bats (A and B; "coping") were able to cope with all three noise types, as their discrimination performance was not affected by noise. In contrast, the other two bats (C and D; "non-coping") were not able to cope with the noise, yet in different ways. Bat C had decreased discrimination performance in all three noise types, took longer in sparse-overlapping noise, and aborted more trials in smooth-overlapping and smooth non-overlapping noise. Bat D had strongly reduced discrimination performance in both smooth and sparse overlapping noise types (but not in non-overlapping noise), made faster decisions in smooth-overlapping treatments, and aborted more trials only in sparse-overlapping noise. Of the changes in echolocation call parameters, the increase in call level was the most prominent one, and shown by both coping and non-coping bats in response to both overlapping noise types. Changes in call frequency were much smaller and without a clear pattern, while call duration increased slightly more for the coping than the non-coping bats. Based on our predictions, both perceptual mechanisms tested,

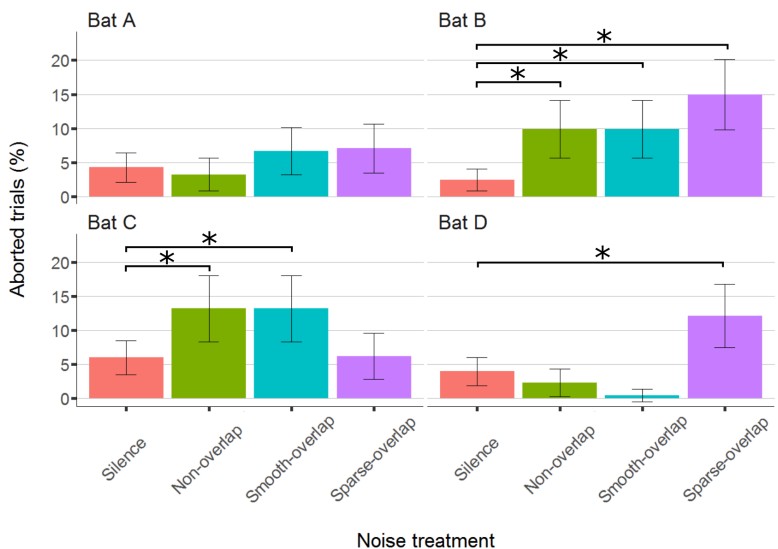

**Figure 5 Percentage of aborted trials during the discrimination task by noise treatment.** Both bats B and C aborted significantly more trials for both non-overlapping and overlapping noise. Yet, bats B and D aborted significantly more trials for sparse-overlapping noise. Box plots show median (solid line) and first and third quartiles (edges of box); whiskers extend to the rest of the data minus outliers (more than 1.5 times the interquartile range), which are shown as points. Noise treatments have been abbreviated here (as compared to the text) to reduce visual clutter (smooth non-overlapping noise = "Non-overlap"; smooth-overlapping noise = "Smooth-overlap"; sparse-overlapping noise = "Sparse-overlap"). Asterisks denote significant differences ($p < 0.05$) in the percentage of aborted trials relative to the silence control.

masking and distraction, appeared to contribute to the bats' performance. In the following, we will discuss all measured parameters in relation to our predictions about the perceptual mechanisms of noise disturbance.

### Discrimination performance (masking vs distraction)

We analyzed the ripple discrimination performance to address the perceptual mechanisms of masking and distraction. Masking should only reduce the performance in overlapping noise, and more so for smooth than sparse overlapping noise. In contrast, distraction should reduce the performance in all noise types, and most so for sparse overlapping noise. Overall, our results do not match those predictions: the coping bats (A and B) showed no decreased performance in any of the noise treatments, excluding masking and distraction. Bat C seemed to suffer from distraction, as its discrimination performance was affected by all noise types. In contrast to our prediction, however, sparse overlapping noise did not reduce performance more than the other noises. Lastly, bat D seemed to suffer from masking, as, in line with our prediction, its discrimination performance was only reduced in both overlapping noise types—yet again without difference between the smooth and sparse noise (in contrast to our prediction). The sparse noise had temporal gaps with a mean duration of 0.3 ms (range: 0–0.6 ms), which is slightly shorter than the average *P. discolor* call here (0.43 ms in silence). Although the detection performance of the gleaning bat *Megaderma lyra* for rustling sounds improved at around this gap duration

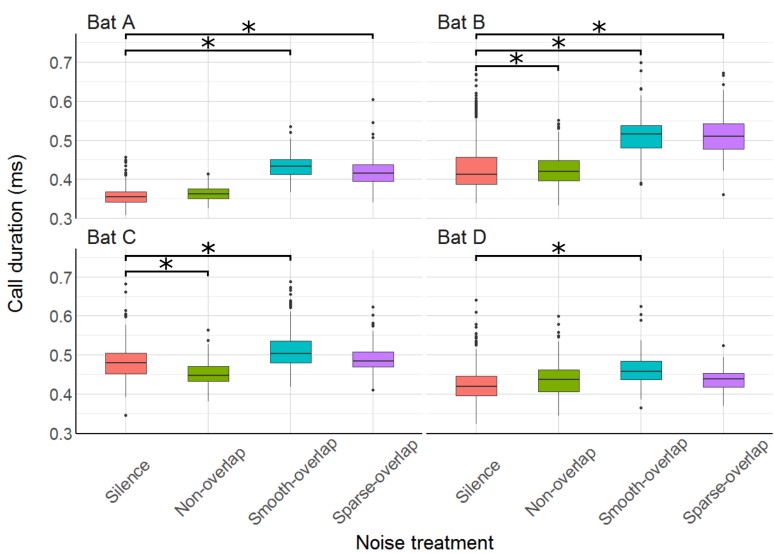

**Figure 6  Duration of echolocation calls during the discrimination tasks by noise treatment.** All bats increased call duration under smooth-overlapping noise, and bats A and B also increased call duration under sparse-overlapping noise. Box plots show median (solid line) and first and third quartiles (edges of box); whiskers extend to the rest of the data minus outliers (more than 1.5 times the interquartile range), which are shown as points. Noise treatments have been abbreviated here (as compared to the text) to reduce visual clutter (smooth non-overlapping noise = "Non-overlap"; smooth-overlapping noise = "Smooth-overlap"; sparse-overlapping noise = "Sparse-overlap"). Asterisks indicate significant differences ($p < 0.05$) of call duration relative to the silence control.

(*Hübner & Wiegrebe, 2003*), it is possible that the temporal gaps in the sparse noise were not sufficiently long to provide sufficient release from masking for echo detection in our species *P. discolor*, despite similarly short call durations of <1 ms in confined spaces (*Goerlitz, Hübner & Wiegrebe, 2008*; *Schuchmann, Hübner & Wiegrebe, 2006*). Therefore, our prediction that sparse-overlapping noise would allow bats to listen in between the gaps of the noise may be false, and further tests with larger gap widths are required. It is also possible that any release from masking that bats had gained might have been opposed by an additional distracting effect of the sparse-overlapping noise opposes, but this seems less likely than the lack of release from masking.

## Trial duration (masking vs distraction)

To further differentiate between masking and distraction as perceptual mechanisms, we also analyzed trial duration as a proxy for task difficulty. Only bat C showed a change in line with our predictions, namely a 26% increase in trial duration in sparse overlapping noise, indicative of stronger distraction by this temporally fluctuating noise. This matches our previous interpretation of this bat's discrimination performance, suggesting that this bat was mostly affected by distraction, which should be strongest for the sparse noise. In contrast, the trial durations in smooth-overlapping noise of both the coping bat A and the non-coping bat D was even shorter than in silence, by 13 and 18%, respectively. In the coping bat A, this faster decision making did not reduce the discrimination performance,

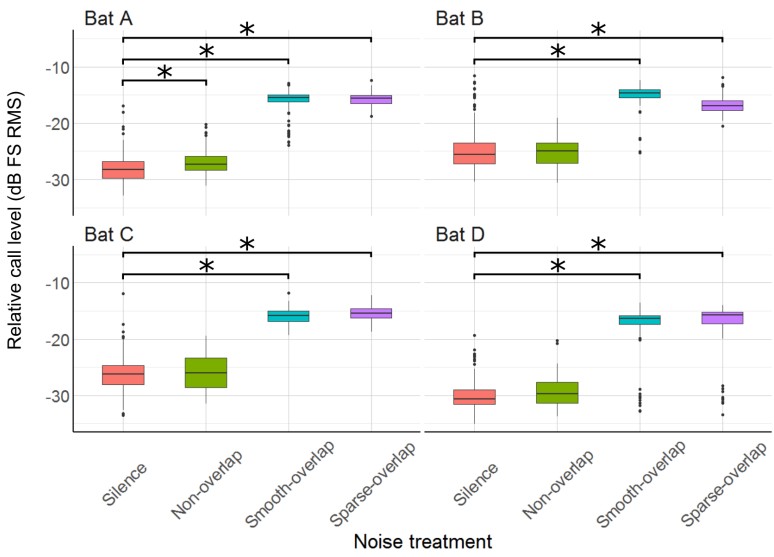

**Figure 7** **Relative sound pressure level of echolocation calls during the discrimination tasks by noise treatments.** All four bats significantly increased call level under both smooth- and sparse-overlapping noise. Only bat A also increased call level under smooth non-overlapping noise, and this change was much smaller. Box plots show median (solid line) and first and third quartiles (edges of box); whiskers extend to the rest of the data minus outliers (more than 1.5 times the interquartile range), which are shown as points. Noise treatments have been abbreviated here (as compared to the text) to reduce visual clutter (smooth non-overlapping noise = "Non-overlap"; smooth-overlapping noise = "Smooth-overlap"; sparse-overlapping noise = "Sparse-overlap"). Asterisks denote significant differences ($p < 0.05$) relative to the silence control.

thus rather indicating reduced task difficulty due to the smooth overlapping noise, which however seems unlikely. In the non-coping bat D, the shorter trial duration might indicate the bat has stopped attempting to complete the task accurately, due to the increased task difficulty by the smooth overlapping noise. This matches our previous interpretation that this bat was affected by masking. However, it is unclear why this bat had equally reduced discrimination performance in sparse overlapping noise, but trial duration was not affected. In summary, trial duration partially supports distraction and masking as perceptual mechanisms of noise disturbance for bats C and D, respectively, but this evidence is not conclusive.

## Echolocation call characteristics (masking)

Several bat species change echolocation call parameters in response to noise (*Bunkley et al., 2015*; *Hage et al., 2013*; *Luo et al., 2017*; *Tressler & Smotherman, 2009*), which is a potential mechanism to mitigate masking effects of noise (*Brumm, 2013*). Thus, we next discuss whether the differences in coping ability (discrimination performance) can be explained by changes in echolocation call parameters. The most prominent change was an increase in call level by around 10–13 dB, shown by all four bats (coping and non-coping) in both overlapping noise types (smooth and sparse). This Lombard effect, the increase of vocalization amplitude in response to noise, is found in many animals from birds to humans (*Brumm & Zollinger, 2011*). Our species, *Phyllostomus discolor*, also exhibits an

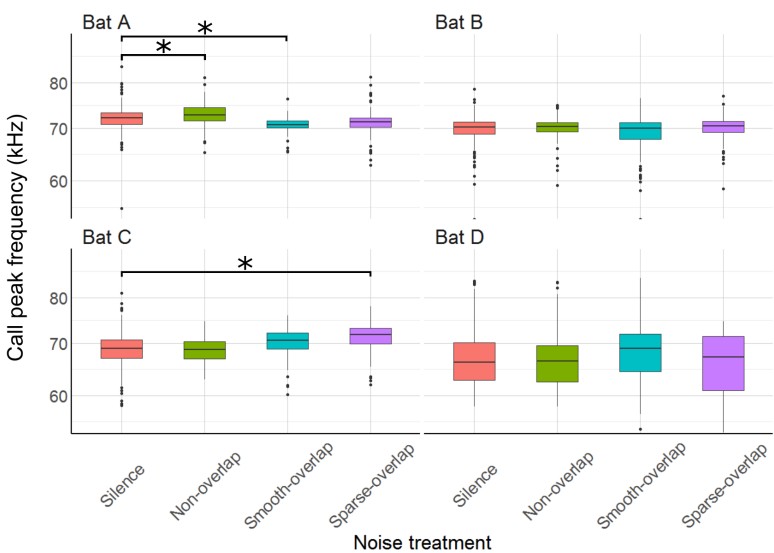

**Figure 8** **Peak frequency of echolocation calls during the discrimination tasks by noise treatment.** Peak frequency of bat A increased under smooth non-overlapping noise and decreased under smooth overlapping noise. Peak frequency of bat C increased under sparse-overlapping noise. Box plots show median (solid line) and first and third quartiles (edges of box); whiskers extend to the rest of the data minus outliers (more than 1.5 times the interquartile range), which are shown as points. Noise treatments have been abbreviated here (as compared to the text) to reduce visual clutter (smooth non-overlapping noise = "Non-overlap" here; smooth-overlapping noise = "Smooth-overlap" here; sparse-overlapping noise = "Sparse-overlap" here). Asterisks denote significant differences ($p < 0.05$) relative to the control treatment.

increasing Lombard effect with increasing noise level, amounting to on average +4 dB for overlapping (40–90 kHz) noise with a level of 52 dB SPL (*Luo et al., 2015*). Here, we show that the Lombard effect increases even further up to 10–13 dB when noise levels are higher (70 dB SPL). This increase in call level is likely a direct response to masking (*c.f.* Fig. 3 *Brumm & Todt, 2002*), as only one of the bats (bat A) increased call amplitude in non-masking noise, and this effect was an order of magnitude smaller (+1.5 dB, c.f. *Luo et al., 2015*). Interestingly, however, although the reaction in call level was equal across all four bats, only two bats (A and B) were able to cope with masking overlapping noise in the discrimination task, while the other two bats (C and D) showed strongly reduced discrimination performance. If we assume that the increased call amplitude provides equal release from masking for all four bats, another perceptual mechanism instead of masking must be responsible for the reduced discrimination performance of the non-coping bats.

In addition to increasing call level, increased call duration improves signal detection in noise because the mammalian ear integrates sound intensity over time (*Heil & Neubauer, 2003*). Indeed, bats increased call duration under noise both in laboratory (*Luo et al., 2015*) and field environments (*Bunkley et al., 2015*). Here, our bats also increased call duration, and did so only in overlapping noise types, suggesting that this was a direct response to masking. We found some differences between coping and non-coping bats. While the coping bats increased call duration by 14–16% in both overlapping noise types (smooth and

sparse), the non-coping bats increased their call duration only in the smooth overlapping noise, and only by 9%. At first view, these patterns are consistent that coping bats mitigate noise masking by increasing call duration, while non-coping bats fail to do so. However, the rather small increase in call duration found here improves signal detectability by only about 1 dB (assuming a gain of 6 dB per doubling of call duration; *Luo et al., 2015*). This is much less than the direct increase in call level (10–13 dB) shown by both coping and non-coping bats, making it unlikely that the slight differences in call duration change can explain the differences in discrimination ability.

Shifting call frequency away from the frequency of a masker is another perceptual mechanism to improve signal detection by reducing spectral overlap, shown by bats when foraging in crowded situations (*Bates, Stamper & Simmons, 2008*; *Gillam, Ulanovsky & McCracken, 2007*; *Ratcliffe et al., 2004*) or near loud ultrasonic insect choruses (*Gillam & McCracken, 2007*). In lower-frequency (5–35 kHz) non-overlapping noise, bat A indeed showed frequency changes consistent with avoiding spectral overlap by increasing its call peak frequency by 1.2 kHz. In contrast, the decrease of its peak frequency around 70 kHz by 1.4 kHz in the higher frequency (40–90 kHz) smooth-overlapping noise is unlikely to improve signal detectability; and correspondingly this bat did not change its peak frequency in the other overlapping noise type (sparse). Bat C increased peak frequency in sparse-overlapping noise only; and the bats B and D showed no response. It is unlikely that such small ($\leq$ 2 kHz) changes in frequency have large effects on call detectability in noise, and thus do not seem insightful for making predictions on the ability of bats to cope with noise.

## Aborted trials

Lastly, the bats avoided the noise types differently. While the coping bat A did not abort more trials under any noise type compared to silence, the other coping bat B aborted more trials in all three noise types (6.0, 4.9, and 11.8 times more in smooth non-overlapping, smooth-overlapping, and sparse-overlapping noise, respectively). This pattern is suggestive of the noise being interpreted as novel or dangerous, perhaps causing fear (i.e., misleading), since the noise type did not affect the discrimination performance and trial duration in this bat (which we would expect if the bat was masked or distracted). The two non-coping bats showed opposite patterns in the number of aborted trials. Bat C aborted more trials in both smooth noise types (2.4 and 5.9 times more in smooth non-overlapping and smooth overlapping noise, respectively), but not in sparse overlapping noise. The response of bat C might indicate that smooth noise types might be more misleading or fear-inducing, as it cannot be linked to masking or distraction. In contrast, bat D aborted more trials in the sparse overlapping noise only (3.9 times more), but not in the two smooth noise types. It is possible here that the sparse overlapping noise was more distracting than the smooth overlapping noise, causing more trials to be aborted (somehow without affecting discrimination performance).

## CONCLUSION

Understanding how echolocating bats deal with noise pollution will be important for their conservation. Additionally, such a mechanistic understanding may also inform the design of noise-based deterrence systems to protect bats from being killed by wind turbines (*Arnett et al., 2013*). Recent studies have shown that echolocating bats avoid noise in the field and lab, when it is possible (*Bunkley et al., 2015*; *Luo, Siemers & Koselj, 2015*; *Schaub, Ostwald & Siemers, 2008*), but as noise sources expand and foraging habitat shrinks, avoidance will become more difficult. Here, when avoidance is impossible, we show that the effects of noise and the underlying perceptual mechanism of disturbance differ at the individual level. It is likely that masking affected all bats, as all of them strongly increased their call levels. However, only two of four bats were able to maintain discrimination performance in noise. Therefore, other perceptual mechanisms, in addition to masking, likely affect signal perception by bats in noise, and probably to different extents for each individual.

By grouping all individuals of one species, we may miss important differences in how individuals deal with noise (reviewed in *Harding et al., 2019*). By ignoring variation across individuals, we may be missing the potential for rapid evolution to occur in response to anthropogenic changes (*Sih, Ferrari & Harris, 2011*). Noise (or other sensory pollutants) can filter individuals over time by selecting for individuals that can cope with noise. Understanding the variation in the ability to cope with noise is paramount to predicting which species may adapt well to encroaching urbanization, and which will not. It is possible that this variation is maintained in natural systems by individual microhabitat selection, because although natural noise is ubiquitous in nature, it is spatially and temporally heterogeneous across the landscape.

## ACKNOWLEDGEMENTS

We would like to thank A. Leonie Baier for help building the experimental setup and training the bats, Karin 'Reni' Heckel for assistance with animal care, Yossi Yovel and Stefan Greif for help measuring light levels, and Jesse R. Barber, Henrik Brumm and two anonymous reviewers for comments on earlier versions of the manuscript (see *Gomes & Goerlitz, 2020a*). We dedicate this research to Lutz Wiegrebe, who passed away in November 2019. Lutz was a great friend and a brilliant scientist with a sharp mind. We will miss his thoughts and jokes forever.

### Funding

This work was supported by the Max Planck Institute for Ornithology, Seewiesen, the Fulbright Program, the National Science Foundation (GRFP 2018268606 to Dylan G.E. Gomes) and Deutsche Forschungsgemeinschaft (DFG, German Research Foundation, Emmy Noether grant 241711556 to Holger R. Goerlitz). The funders had no role in study design, data collection and analysis, decision to publish, or preparation of the manuscript.

### Grant Disclosures

The following grant information was disclosed by the authors:

Max Planck Institute for Ornithology, Seewiesen.

National Science Foundation: GRFP 2018268606.

Deutsche Forschungsgemeinschaft (DFG, German Research Foundation, Emmy Noether): 241711556.

### Competing Interests

The authors declare there are no competing interests.

### Author Contributions

- Dylan G.E. Gomes conceived and designed the experiments, performed the experiments, analyzed the data, prepared figures and/or tables, authored or reviewed drafts of the paper, and approved the final draft.
- Holger R. Goerlitz conceived and designed the experiments, analyzed the data, authored or reviewed drafts of the paper, and approved the final draft.

### Animal Ethics

The following information was supplied relating to ethical approvals (i.e., approving body and any reference numbers):

Bat housing and all research was approved by the German authorities under the permit numbers 311.5-5682.1/1-2014-023 (Landratsamt Starnberg) and 55.2-1-54-2532-18-15 (Regierung von Oberbayern), respectively.

### Data Availability

All data and code are available on Zenodo:

Dylan Gomes, & Holger Goerlitz. (2020). Data and code for: Individual differences show that only some bats can cope with noise-induced masking and distraction. http://doi.org/10.5281/zenodo.3928601.

### Supplemental Information

Supplemental information for this article can be found online at http://dx.doi.org/10.7717/peerj.10551#supplemental-information.

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

## PeerJ

**Gomes DG, Goerlitz HR. 2020.** Individual differences show that only some bats can cope with noise-induced masking and distraction. *bioRxiv preprint* DOI 10.1101/2020.07.04.188086.

**Gomes DG, Halfwerk W, Taylor RC, Ryan MJ, Page RA. 2017.** Multimodal weighting differences by bats and their prey: probing natural selection pressures on sexually selected traits. *Animal Behavior* **134**:99–102 DOI 10.1016/j.anbehav.2017.10.011.

**Gomes DG, Page RA, Geipel I, Taylor RC, Ryan MJ, Halfwerk W. 2016.** Bats perceptually weight prey cues across sensory systems when hunting in noise. *Science* **353**:1277–1280 DOI 10.1126/science.aaf7934.

**Grunwald J-E, Schörnich S, Wiegrebe L. 2004.** Classification of natural textures in echolocation. *Proceedings of the National Academy of Sciences of the United States of America* **101**:5670–5674 DOI 10.1073/pnas.0308029101.

**Hage SR, Jiang T, Berquist SW, Feng J, Metzner W. 2013.** Ambient noise induces independent shifts in call frequency and amplitude within the Lombard effect in echolocating bats. *Proceedings of the National Academy of Sciences of the United States of America* **110**:4063–4068 DOI 10.1073/pnas.1211533110.

**Halfwerk W, Holleman LJ, Slabbekoorn H. 2011.** Negative impact of traffic noise on avian reproductive success. *Journal of Applied Ecology* **48**:210–219.

**Harding HR, Gordon TA, Eastcott E, Simpson SD, Radford AN. 2019.** Causes and consequences of intraspecific variation in animal responses to anthropogenic noise. *Behavioral Ecology* **30**:1501–1511 DOI 10.1093/beheco/arz114.

**Harrison XA, Donaldson L, Correa-Cano ME, Evans J, Fisher DN, Goodwin CE, Robinson BS, Hodgson DJ, Inger R. 2018.** A brief introduction to mixed effects modelling and multi-model inference in ecology. *PeerJ* **6**:e4794 DOI 10.7717/peerj.4794.

**Hartmann WM, Pumplin J. 1988.** Noise power fluctuations and the masking of sine signals. *Journal of the Acoustical Society of America* **83**:2277–2289 DOI 10.1121/1.396358.

**Heil P, Neubauer H. 2003.** A unifying basis of auditory thresholds based on temporal summation. *Proceedings of the National Academy of Sciences of the United States of America* **100**:6151–6156 DOI 10.1073/pnas.1030017100.

**Hoffmann S, Baier L, Borina F, Schuller G, Wiegrebe L, Firzlaff U. 2008.** Psychophysical and neurophysiological hearing thresholds in the bat Phyllostomus discolor. *Journal of Comparative Physiology. A, Sensory, Neural, and Behavioral Physiology* **194**:39–47 DOI 10.1007/s00359-007-0288-9.

**Holderied MW, Korine C, Fenton MB, Parsons S, Robson S, Jones G. 2005.** Echolocation call intensity in the aerial hawking bat Eptesicus bottae (Vespertilionidae) studied using stereo videogrammetry. *Journal of Experimental Biology* **208**:1321–1327 DOI 10.1242/jeb.01528.

**Hübner M, Wiegrebe L. 2003.** The effect of temporal structure on rustling-sound detection in the gleaning bat, Megaderma lyra. *Journal of Comparative Physiology. A, Sensory, Neural, and Behavioral Physiology* **189**:337–346 DOI 10.1007/s00359-003-0407-1.

**Kjellberg A, Landström ULF, Tesarz M, Söderberg L, Akerlund E. 1996.** The effects of nonphysical noise characteristics, ongoing task and noise sensitivity on annoyance and distraction due to noise at work. *Journal of Environmental Psychology* **16**:123–136 DOI 10.1006/jevp.1996.0010.

**Knoblauch K. 2007.** psyphy: functions for analyzing psychophysical data in R. R package version 00-5. URL HttpCRAN R-Proj. Orgpackage Psyphy.

**Kwiecinski GG. 2006.** Phyllostomus discolor. *Mammalian Species* **2006**:1–11.

**Lattenkamp EZ, Vernes SC, Wiegrebe L. 2018.** Volitional control of social vocalisations and vocal usage learning in bats. *Journal of Experimental Biology* **221**:1–8.

**Lazure L, Fenton MB. 2011.** High duty cycle echolocation and prey detection by bats. *Journal of Experimental Biology* **214**:1131–1137 DOI 10.1242/jeb.048967.

**Luo J, Goerlitz HR, Brumm H, Wiegrebe L. 2015.** Linking the sender to the receiver: vocal adjustments by bats to maintain signal detection in noise. *Scientific Reports* **5**:18556.

**Luo J, Lingner A, Firzlaff U, Wiegrebe L. 2017.** The Lombard effect emerges early in young bats: implications for the development of audio-vocal integration. *Journal of Experimental Biology* **220**:1032–1037 DOI 10.1242/jeb.151050.

**Luo J, Siemers BM, Koselj K. 2015.** How anthropogenic noise affects foraging. *Global Change Biology* **21**:3278–3289 DOI 10.1111/gcb.12997.

**Luther D, Baptista L. 2009.** Urban noise and the cultural evolution of bird songs. *Proceedings of the Royal Society B: Biological Sciences* **277**:469–473.

**Matthews KA, Scheier MF, Brunson BI, Carducci B. 1980.** Attention, unpredictability, and reports of physical symptoms: eliminating the benefits of predictability. *Journal of Personality and Social Psychology* **38**:525 DOI 10.1037/0022-3514.38.3.525.

**Morris-Drake A, Kern JM, Radford AN. 2016.** Cross-modal impacts of anthropogenic noise on information use. *Current Biology* **26**:R911–R912 DOI 10.1016/j.cub.2016.08.064.

**Naguib M, Van Oers K, Braakhuis A, Griffioen M, De Goede P, Waas JR. 2013.** Noise annoys: effects of noise on breeding great tits depend on personality but not on noise characteristics. *Animal Behavior* **85**:949–956 DOI 10.1016/j.anbehav.2013.02.015.

**Nemeth E, Brumm H. 2010.** Birds and anthropogenic noise: are urban songs adaptive? *The American Naturalist* **176**:465–475 DOI 10.1086/656275.

**Purser J, Radford AN. 2011.** Acoustic noise induces attention shifts and reduces foraging performance in three-spined sticklebacks (Gasterosteus aculeatus). *PLOS ONE* **6**:e17478 DOI 10.1371/journal.pone.0017478.

**R Core Team. 2017.** R: a language and environment for statistical computing. Vienna: R Foundation for Statistical Computing. *Available at https://www.r-project.org/*.

**Rabin LA, McCowan B, Hooper SL, Owings DH. 2003.** Anthropogenic noise and its effect on animal communication: an interface between comparative psychology and conservation biology. *International Journal of Comparative Psychology* **16**:172–192.

**Ratcliffe JM, Hofstede HMter, Avila-Flores R, Fenton MB, McCracken GF, Biscardi S, Blasko J, Gillam E, Orprecio J, Spanjer G. 2004.** Conspecifics influence call design

in the Brazilian free-tailed bat, Tadarida brasiliensis. *Canadian Journal of Zoology* **82**:966–971 DOI 10.1139/z04-074.

**Schaub A, Ostwald J, Siemers BM. 2008.** Foraging bats avoid noise. *Journal of Experimental Biology* **211**:3174–3180 DOI 10.1242/jeb.022863.

**Schuchmann M, Hübner M, Wiegrebe L. 2006.** The absence of spatial echo suppression in the echolocating bats Megaderma lyra and Phyllostomus discolor. *Journal of Experimental Biology* **209**:152–157 DOI 10.1242/jeb.01975.

**Siemers BM, Schaub A. 2011.** Hunting at the highway: traffic noise reduces foraging efficiency in acoustic predators. *Proc. R. Soc. Lond. B Biol. Sci* **278**:1646–1652.

**Sierro J, Schloesing E, Pavón I, Gil D. 2017.** European blackbirds exposed to aircraft noise advance their chorus, modify their song and spend more time singing. *Frontiers in Ecology and Evolution* **5**:68 DOI 10.3389/fevo.2017.00068.

**Sih A, Ferrari MC, Harris DJ. 2011.** Evolution and behavioural responses to human-induced rapid environmental change. *Evolutionary Applications* **4**:367–387 DOI 10.1111/j.1752-4571.2010.00166.x.

**Simmons AM, Ertman A, Hom KN, Simmons JA. 2018.** Big brown bats (Eptesicus fuscus) successfully navigate through clutter after exposure to intense band-limited sound. *Scientific Reports* **8**:1–13 DOI 10.1038/s41598-017-17765-5.

**Simpson SD, Radford AN, Nedelec SL, Ferrari MC, Chivers DP, McCormick MI, Meekan MG. 2016.** Anthropogenic noise increases fish mortality by predation. *Nature Communications* **7**:10544.

**Standing L, Lynn D, Moxness K. 1990.** Effects of noise upon introverts and extroverts. *Bulletin of the Psychonomic Society* **28**:138–140 DOI 10.3758/BF03333987.

**Tanner Jr WP. 1958.** What is masking? *Journal of the Acoustical Society of America* **30**:919–921 DOI 10.1121/1.1909406.

**Tressler J, Smotherman MS. 2009.** Context-dependent effects of noise on echolocation pulse characteristics in free-tailed bats. *Journal of Comparative Physiology. A, Sensory, Neural, and Behavioral Physiology* **195**:923–934 DOI 10.1007/s00359-009-0468-x.

**Tyack PL, Zimmer WM, Moretti D, Southall BL, Claridge DE, Durban JW, Clark CW, D'Amico A, DiMarzio N, Jarvis S. 2011.** Beaked whales respond to simulated and actual navy sonar. *PLOS ONE* **6**:e17009 DOI 10.1371/journal.pone.0017009.

**Vélez A, Bee MA. 2011.** Dip listening and the cocktail party problem in grey treefrogs: signal recognition in temporally fluctuating noise. *Animal Behavior* **82**:1319–1327 DOI 10.1016/j.anbehav.2011.09.015.

**Voellmy IK, Purser J, Flynn D, Kennedy P, Simpson SD, Radford AN. 2014.** Acoustic noise reduces foraging success in two sympatric fish species via different mechanisms. *Animal Behavior* **89**:191–198 DOI 10.1016/j.anbehav.2013.12.029.

**West-Eberhard MJ. 1984.** Sexual selection, competitive communication and species specific signals in insects. In: *Insect communication (Proceedings of the 12th symposium of the Royal Entomological Society of London.* Cambridge, MA, USA: Acedemic Press.

**Wiley RH. 2013.** Signal detection, noise, and the evolution of communication. In: *Animal communication and noise.* Berlin, Germany: Springer, 7–30.

**Zhou Y, Radford AN, Magrath RD. 2019.** Why does noise reduce response to alarm calls? Experimental assessment of masking, distraction and greater vigilance in wild birds. *Functional Ecology* **33**:1280–1289 DOI 10.1111/1365-2435.13333.