# Peer review of "Individual differences show that only some bats can cope with noise-induced masking and distraction"

_PeerJ, doi:10.7717/peerj.10551_

## Round 0.1 · original submission · Minor Revisions

Please pay close attention to the comments of both reviewers. I am sorry for the delay, but I was waiting for a 3rd review which never came in.

Reviewer 1 ·

Basic reporting

-The manuscript is generally well written and the authors are generally clear in communicating their thoughts. However, throughout the manuscript there are some run on sentences that are long winded that I believe would be clearer if broken into multiple sentences, as well as a few points where grammar needs to be check or more clarity added. As such the manuscript should be carefully edited. Some examples of these occur at lines:
• Line 47-51: provides examples of where individual differences are documented than later states “this” is often overlooked. Clearly state “this” as individual differences. And I assume this is true although you just provided a handful of citations contradictory to this statement?
• Line 74: more clearly state what is meant by “tracking efforts and sensory strategies”
• Line 74: unclear at first if author is trying to improve bat perception relatively easy, or how bats improve their perception with ease.
• Line 75: I feel like this line clearly states what is in the sentence before. Just state what you mean, no need to ramble for multiple sentences 
• Line 83-83: “increasing difficulty”. Not clear at time how you increased difficulty.
• Line 94-95: State the opposites as opposed to just saying the opposite would be true in masking.

-The introduction briefly describes the background of noise disturbances in animals and quickly gets into the details of auditory mechanisms that may be causing this disruption. Various aspects of noise disruptions (masking, distraction, etc) are well described and hypotheses clearly stated.

-While I appreciate the brevity of the intro, I would like to see more details on some past animal studies with noise.
• I appreciate the general emphasis in intro/discussion on importance of individual differences and not just group means
• Line45 “Yet it is not often understood what mechanisms drive these changes”. In the few studies that do get to understand the mechanisms, what do the show? Or do NONE test mechanisms?
• Line 40: Authors discuss anthropogenic noise. Do any studies compare the frequency distributions of anthropogenic noises in cities, parks, nature areas and at what frequency’s the maximum energy occur. Are these anthropogenic noises within the bats echolocation range? Are there measurements of anthropogenic noise increases in areas over decades or years to show the increase? These may guide the hypothesis of distraction vs masking, and provide more details to the context of anthropogenic noise disturbance to bats in particular.

-Discussion brings up windmills and bat ecology. Add a sentence explaining why this is an issue for non-bat scientists.

-Figures are informative and related to main data discussed in manuscript. I put for the following suggestions for increase clarity:
• Ensure the use of colour-blind friendly colour palette
• I suggest dotted or broken lines, or use of squares/circles/etc so figures are clear for individuals printing in greyscale (yes, those who print still exist). Would solve above issue with colour-blind palette.
• Figure captions are clear but I recommend stating what each graph represents, what boxplots are, and any described patterns by bats prior to stating the asterisks denote significance
• I assume statistical significance in figure descriptions is p<0.05, clearly state
• Fig2 – “Ripple Height” on X-axis appears partially cut off. If legend is to be within an animal panel I would suggest Bat A (top, left panel) as opposed to Bat B.
• Fig1- Is it possible to put a real picture of the apparatus in addition or in lieu of panel A?

-Raw data is provided and data measurements conform to those of the field.

Experimental design

-The authors present an original study with well defined questions, predictions, and aimed to fill a specific gap in the scientific literature. All methods are ethical and had ethics approval.

-The mushroom maze is established (Baier 2019) but not common in the field for perception studies, especially compared to standard y-maze platforms.
• Objective decision measurements are good and give animals room to explore. Did animals ever not trigger the sensor but appear to roost above it and behaviourally make a decisions?
• Consider adding a brief statement regarding echolocating through mesh or possible disruption between mesh/ripple depth when echolocating for behavioural and non-auditory audience

-Authors should include details regarding animals used in the experiment and selection criteria. Considering 4/9 were able to be trained reliably where these individuals older or have previous training for other experiments, from wild populations and not bred in captivity, etc. As a reader I immediately wonder if age or past experience facilitate learning and real world acoustic accommodations vs a juvenile with less experience or born in captivity.

-Authors provide sufficient details regarding experimental methodologies, statistics, etc. as well as ample reasoning as to the methods used. (ex. Statistical modelling can be subjective and the authors include why they deemed factors like length of day in experience to be important).

-Lines 193-194: I understand all trials of one condition were completed prior to the next auditory stimulus likely for ease of experimenter, but over time with the same stimuli could individuals start to “tune out” the smooth/sparse/non-overlapping noise? Similar to how over time humans “tune out” traffic or background noise. Could this impact results? Statement to why this procedural decision was made would be appreciated.

-Lines 199-202: Is it standard to start with easiest discrimination and move to hardest? Could there be animal learning during this time? Was animal previously trained to leave through the exit door is this just innate behaviour where animals tried to escape? What were protocols to determine animal was satiated (ex. 3 consecutive trials without eating the mealworm? Or randomly determined).

-Line 213: clarify whether 4s was recorded prior to decision or if posthoc a longer recording was cut to only include 4s prior to decision.

-Line 238: I do not believe the parentheses describe both highest and lowest frequency

-Excellent details on lighting and past visual aspects of bat vision, as well as how specific factors were decided upon to be placed into the model and why others were not included.

Validity of the findings

-Data are provided, statistics are sound, and experiments test the hypotheses provided.

-Authors are clear when speculating on any results, and clearly state whether their hypotheses are supported.

-Descriptions of results are appropriate

Additional comments

The authors have presented a study that is of importance to understanding bat auditory perception with future ecological relevance. This manuscript read clearly and logically, and the experimental design was well thought out with the appropriate measurements evaluated. It was a pleasure to read this manuscript.

I commend the authors on this extensive and time intense study. The amount of work in this study is huge. Kudos to you! My condolences on the passing of your mentor and friend. I’m sure you have made them proud with the work you have presented here.

Reviewer 2 ·

Basic reporting

The purpose of this manuscript was to investigate how acoustic noise influences behavior and auditory perception in individual echolocating pale spear-nosed bats (Phyllostomus discolor). Bats were tested in the lab and trained on a psychoacoustical discrimination task in a variation of a Y-maze or T-maze that allowed animals to crawl toward a rewarded stimulus (i.e. a smooth disk) in a 2-alternative forced choice (2-AFC) paradigm. In the maze bats were tasked to discriminate the trained rewarded stimulus (flat disk) from an unrewarded stimuli (i.e. one of seven types concentric rippled disks all of the same spatial frequency but with differing ripple amplitudes). During testing, the performance of four trained bats was tested in silence (i.e. no acoustic playback) and with three types of noise via acoustic playback. The results showed that all four bats were able to complete the 2-AFC task but to vary degrees as the psychoacoustic threshold of 70% correct response varied for selecting the smooth disk versus a rippled disk with various ripple amplitudes. In short, the behavioural performance of two bats (A&B) seemed to be largely unaffected by noise playback whereas two other bats (C&D) had much higher (worse) ripple thresholds. As expected, all bats increased the amplitude of their echolocation calls in one or more trials during acoustic playback of noise; however, other echolocation call parameter changes shown by each individual were rather small (i.e. call duration, peak echolocation frequency) and the authors concluded that such differences were unlikely to explain overall differences observed in the behavioural performance of bats A&B compared to bats C&D. Moreover, trial duration and the proportion of aborted trials did not see to correlate with the behavioural performance of the specific four individual bats (out of nine) that managed to complete the study. The manuscript demonstrates there was high individual variability in response to noise treatments but the data do not show a clear pattern with respect to behavior and neurobiological mechanisms that could explain the results.

Experimental design

Overall, I found the science in this manuscript to be interesting and the Methods appropriate. The study was fairly well designed and carefully conducted as was the data analysis. I also found the manuscript to be well-written and easy to read. All of my suggestions below are intended to help improve the manuscript.

Validity of the findings

N/A

Additional comments

SPECIFIC COMMENTS

1. One shortcoming of the manuscript is that at least three important citations were not included. The studies below were also conducted in the laboratory, albeit on a different species of echolocating bat, and demonstrate that some bats are unaffected by exposure to various types of noise. I mention this upfront because the authors seem to be unaware of these reports, which are directly related and highly relevant to their study.

Simmons AM, Hom KN, Warnecke M, Simmons JA (2016). Broadband noise exposure does not affect hearing sensitivity in big brown bats (Eptesicus fuscus). The Journal of Experimental Biology, 219(7):1031-1040.

Simmons AM, Hom KN, Simmons JA (2017) Big brown bats (Eptesicus fuscus) maintain hearing sensitivity after exposure to intense band-limited noise. The Journal of the Acoustical Society of America, 141(3):1481-1489.

Simmons AM, Ertman A, Hom KN, Simmons JA (2018) Big brown bats (Eptesicus fuscus) successfully navigate through clutter after exposure to intense band-limited sound. Scientific Reports 10, 8(1):1-3.


2. (lines 47-51) Individual differences in response to noise has also been document in bats (see point 1).

3. (lines 51-52) I agree with this point; however, I am still unclear as to how the mechanisms of “masking” can be independently dissociated from the other suggested perceptual mechanisms (?) of noise disturbance such as distraction and misleading (also, the term misleading as an independent mechanism from distraction may itself be misleading). For example, both masking and distraction have physiological and attentional components that may not necessarily be mutually exclusive from each other when measured with behavior. This could also be true for stress, fear, and avoidance.

4. (line 70) Major points. This sentence may be misleading. It is my opinion that the manuscript did not tease apart the relative contributions of masking and distraction as perceptual mechanisms because it is unclear that they are mutually exclusive when measured with behavior. (lines 80-81). I agree that when noise does not overlap with the bat’s echolocation calls this may result in less masking of returning echoes but even at “off-biosonar signal frequencies” masking can still occur. Moreover, playbacks of noise with spectrally overlapping and non-overlapping bandwidths (re echolocation calls) can also be attentionally distracting. Increasing task difficulty by using disks with lower ripple amplitudes decreases the available signal-to-noise ratio (SNR) in echoes and thus in the bat’s ability to make a perceptual decision, but it is unclear that this is a mechanism of distraction because when the SNR becomes too low there is no signal for the bat to detect or be distracted by. (lines 86-88) I do not completely understand the logic of this prediction. Distraction could be just as or more prominent for acoustic playbacks that spectrally overlap the bat’s own echolocation calls and thus received echoes regardless of the noise temporal structure. Also, the introduction of temporal gaps in the noise may introduces additional signal frequencies and/or a cue (i.e. they sound rougher… lines 149-150) and opportunities for detection, masking, and distraction (as suggested on lines 88-92). (lines 92-95) It is unclear to me that the predictions are consistent with the background knowledge presented.

5. (lines 182-183) Minor point. Please provide additional clarifying text. Was the entire testing arena on a rotating platform? Why did you say the disks were “swiveled” to their positions?

6. (lines 194-195) Perhaps I missed it, but I am unclear on this point because I did not see an explicit comparison of the bat’s performance in silence for the first test and then the re-test condition in silence. What was done to assure the reader that performance was not due to learning or other effects?

7. (lines 207-208) The manuscript uses the number of aborted trials as a metric to indicate the bat’s aversion to noise but this is not necessarily a good measure because all four bats aborted some trials even in the silence condition with no noise playback. At least two bats had, on average, lower numbers of abortions to a noise treatment than the silence treatment. My point is that it is very difficult to know why a bat aborts any trial and in at least two animals it appears that trials were aborted for reasons other than aversion to noise.

8. (lines 219-232) I am wondering if the small changes in signal duration and spectral frequency may have more to do with the impressive methods used to extraction signals from within noise and delineate the extracted calls? I don’t know. Were the same methods used to extract and delineate calls in the silence condition?

9. (lines 244-246) This would seem to make sense… analyze the loudest call recorded by one of the two microphones. When a call was detected on both microphones, presumably it was always louder in the microphone nearest to where the bat broke the infrared beam (i.e. the decision point)? Is this true? How often did you measure calls recorded from the microphone that was located further away and opposite to the side where the bat broke the IR beam (presumably, this did not happen very often).

10. (lines 254-256) Minor point. Consider moving these results to the Results section.

11. (lines 293-295) Because the Methods made a point of saying that the number of days that a bat was in an experiment may be an important variable to analyze, what were the Results from this analysis? Did you find that your bats learned the task or that they took longer to complete trials that you thought would be more difficult tasks?

12. (lines 326-329) Minor point. I’m not sure the terms “coping” and “non-coping” are the best to use to describe bats A&B from C&D. All bats had to cope with the experimental treatments. Maybe consider alternative terms and something a little less anthropomorphic?

13. (lines 395-397; lines 402-404) I agree because I think it is very difficult to design a task that mutually excludes mechanisms of distraction from those of masking.

14. (lines 426-428) An increase in trial duration could also be predicted under a mechanism of masking.

15. (lines 450-455) What is the evidence that the increase in echolocation call amplitude was involuntary? (lines 461-463) An increase in call amplitude should also result in less distraction because the amplitude of the received echo (i.e. the SNR) should also be louder.

---

## Round 0.2 · accepted · Accept

I appreciate your efforts to address the reviewers' comments.